# The Making of Transgenic *Drosophila guttifera*

**DOI:** 10.3390/mps3020031

**Published:** 2020-04-27

**Authors:** Mujeeb Shittu, Tessa Steenwinkel, Shigeyuki Koshikawa, Thomas Werner

**Affiliations:** 1Department of Biological Sciences, Michigan Technological University, Houghton, MI 49931, USA; mshittu@mtu.edu (M.S.); testeenw@mtu.edu (T.S.); 2Faculty of Environmental Earth Science, Hokkaido University, N10W5, Kita-ku, Sapporo, Hokkaido 060-0810, Japan; 3Graduate School of Environmental Science, Hokkaido University, N10W5, Kita-ku, Sapporo, Hokkaido 060-0810, Japan

**Keywords:** microinjection, *Drosophila guttifera*, transgenesis, *piggyBac*

## Abstract

The complex color patterns on the wings and body of *Drosophila guttifera* (*D. guttifera*) are emerging as model systems for studying evolutionary and developmental processes. Studies regarding these processes depend on overexpression and downregulation of developmental genes, which ultimately rely upon an effective transgenic system. Methods describing transgenesis in *Drosophila melanogaster* (*D. melanogaster*) have been reported in several studies, but they cannot be applied to *D. guttifera* due to the low egg production rate and the delicacy of the eggs. In this protocol, we describe extensively a comprehensive method used for generating transgenic *D. guttifera*. Using the protocol described here, we are able to establish transgenic lines, identifiable by the expression of enhanced green fluorescent protein (EGFP) in the eye disks of *D. guttifera* larvae. The entire procedure, from injection to screening for transgenic larvae, can be completed in approximately 30 days and should be relatively easy to adapt to other non-model *Drosophila* species, for which no white-eyed mutants exist.

## 1. Introduction

Our ability to genetically manipulate any organism of choice has become a powerful tool for studying gene function. The introduction of a foreign DNA into the genome of an organism to produce germline transformations is known as transgenesis. Until recently, genetic manipulation in *Drosophila* has been largely achieved by transposase-mediated transgenesis [1,2]. In this technique, transposases catalyze the integration of a transposon into the genome of an organism in an unpredictable manner [3]. However, in recent years, significant improvements in fly genetic techniques have been reported, particularly the integration of transgenes into the genome in a site-specific manner through the use of different integrases and recombinases [4,5,6,7]. These techniques have been used to manipulate a variety of species [7,8,9], but arguably, most of these studies have been conducted in *D. melanogaster*. Many non-model *Drosophila* species offer a variety of life history and morphological traits that are absent in *D. melanogaster* [10]. Thus, our transgenic protocol will enable researchers to study new phenomena beyond the commonly used model organism.

*Drosophila guttifera* is a rare mycophagous species in the quinaria species group, native to the Midwest and Southeast of the USA [10]. *D. guttifera* has recently become a useful model for pigmentation studies [11]. Most studies carried out on this species are focused on unraveling the functional mechanism for the formation of polka-dotted patterns on the wings of this species. For example, Werner et al. (2010) reported that the wing spots are induced by the Wingless morphogen, and the unique polka-dotted expression pattern of *wingless* is due to the evolution in *cis*-regulatory elements of the *wingless* gene [12,13,14]. Furthermore, Fukutomi et al. (2017) conducted a study measuring the timing of melanin deposition in wings, using a transgenic *D. guttifera* line [15]. Ongoing and future studies in the field of color pattern development depend on the overexpression and downregulation of genes, which ultimately rely on an effective and efficient transgenesis system.

Previously, a review highlighting strategies for making transgenic *D. melanogaster* was published [16]. However, this protocol cannot be directly applied to *D. guttifera* transgenesis for the following reasons: (1) The egg production rate of *D. guttifera* is one to two orders of magnitude lower than that of *D. melanogaster*, and thus, a large fly population is needed; (2) *D. guttifera* flies die quickly from the ethanol fumes produced by the yeast required for egg-laying, requiring proper ventilation during the egg collection process; (3) *D. guttifera* females stick their eggs into the substrate, making them virtually impossible to collect from an agar surface; (4) *D. guttifera*’s eggs are covered by a thin proteinaceous chorion, which causes rapid embryonic desiccation in this species; (5) there is no white-eyed mutant available for *D. guttifera*, which is why larvae have to be screened for fluorescent markers; and (6) besides the *piggyBac* transposon, most other tested transposons do not result in transgenic *D. guttifera* [12,14,15,17]. For these reasons, critical steps must be deployed to successfully transform *D. guttifera*.

Here, we provide a detailed approach for creating transgenic *D. guttifera*. This protocol is accompanied by a video (Appendix A) and troubleshooting steps (Table 1), which makes it easy for researchers in the field to follow. To the best of our knowledge, this protocol is the first to describe how to make transgenic flies in a non-model *Drosophila* species. It is our expectation that this protocol can be adopted by researchers who might be interested in experiments involving transgenesis in other non-model insect species.

## 2. Experimental Design

### 2.1. Required Materials and Equipment

#### 2.1.1. Materials

Plexiglas egg-laying cage measuring 300 mm × 200 mm × 200 mmLampInstant yeast (Instaferm)SpongesMedium-sized Petri dishes (100 mm × 15 mm)1-L plastic beakers for wet chambers2-L plastic jar5-L beakerEgg collection filter with Corning Netwell insert (fine-filter, New York, NY, USA)Egg collection filter with Corning Netwell insert (coarse-filter, New York, NY, USA)Squeeze bottleMicro cover glass (18 mm × 18 mm)Microslide (25 mm × 75 mm, 1 mm thick)Small brush (to move egg masses)Very fine brush (to line up individual eggs)Halocarbon oil 700 (Sigma-Aldrich, cat. no. H8898, Albany, NY, USA)Halocarbon oil 27 (Sigma-Aldrich, cat. no. H8773, Albany, NY, USA)10-mL syringe50-mL Falcon tubeFHC capillary tube (Borosil 1.0 mm × 0.75 mm ID/Fiber with Omega dot fiber)Nitrogen gasCO_2_ gasFly food vialsCornmeal-sucrose-yeast medium (fly food) [10]Aluminum foilTwo pairs of forcepsSpatulaCotton plugsMoist chamberAnti-fungus paper (see Section 5.1)

#### 2.1.2. Equipment

Flaming/Brown micropipette puller Model P-97 (Sutter Instruments, Novato, CS, USA)Microinjector (Narishige IM 300, Amityville, NY, USA)Needle-holder (Narishige, Amityville, NY, USA)Micromanipulator (Narishige, Amityville, NY, USA)Inverted microscope (Olympus CKX31, Center Valley, PA, USA)Dissecting microscopeWater bath (Thermo Scientific, HAAK S3, Waltham, MA, USA)Mercury burner (Olympus U-RFL-T, Center Valley, PA, USA)Imaging microscope (Olympus SZX16, Center Valley, PA, USA) with fluorescence filters for GFP and DsRed

## 3. Methods

### 3.1. The Egg-Laying Cage

The egg-laying cage is made of Plexiglas that allows good aeration through the nylon entrance at the front and the plastic mesh at the back of the cage. Aeration is necessary to prevent the accumulation of toxic ethanol fumes produced by the yeast. The cage measures 300 mm × 200 mm × 200 mm and holds approximately 10,000 flies from 20 bottles. Before a clean cage can be populated, six small (35 mm × 10 mm) plastic Falcon Petri dish bottom halves are taped upside-down (using double-sided tape) onto the bottom of the cage. They will serve as “tables” for the bigger cornmeal-sucrose-yeast medium plates and must be spaced out evenly. Medium-sized empty dishes can be used to see if the spacing is good. Without these “tables”, the cornmeal-sucrose-yeast medium plates would crush lots of flies on the ground with every food exchange (Figure 1A,B).

Feed the flies in the cage every 2–3 days with six cornmeal-sucrose-yeast medium plates at a time, which are medium-sized (100 mm × 15 mm) Petri dish bottom halves, filled to the top with cornmeal-sucrose-yeast medium. Right before feeding the flies, wipe off any excess water from the plates with a paper towel and sprinkle with some dry baker’s yeast. When replacing old food plates with new plates, follow the three rules below to minimize the killing of flies: (1)Knock the plate gently against the round Petri dish “table” to remove any flies before taking out a plate. Use the thumb and middle finger to hold the dish, while holding the index finger in place to prevent the food from falling out.
(a)If the gap between the food and the Petri dish is large due to excessive drying of the food, then bump (knock the flies) towards the large gap. The food will not move much downwards, and no fly will be trapped.(b)If the gap is narrow, bump towards the side where the food still sticks to the plastic. This prevents most of the flies in the gap from being squeezed to death.
(2)One out, one in! Always take only one plate out at a time and replace immediately with a fresh plate. If there are flies on the food “table” in the cage, they can be gently pushed away with the front edge of the fresh food plate when putting it down. Then repeat this step with the remaining plates.(3)Freeze and discard the used food as the flies developing from it will be of poor quality and a source of fungus infestations.

An egg-laying cage can be in operation for up to 10 weeks before the flies must be transferred to a new cage. If fungus grows (pay attention to places where the food comes in contact with the wall), change the cage earlier. Perform the following steps to move the flies to a new cage:(1)Take out all cornmeal-sucrose-yeast medium plates, close them, and continue working at a fly bench with CO_2_ gas.(2)Cover an area with unfolded paper towels to your right and put a 5-L beaker on it.(3)Have a new cage with a fresh entrance ring plus nylon ready.(4)Put the populated cage to the left of the new cage (not on the towels), the entrance facing upwards to prevent CO_2_ from leaking out of the cage through the back wall.(5)Open the CO_2_ gas, lead it through the nylon to anesthetize the flies, and then close the CO_2_ gas supply.(6)Remove the ring with the nylon of the old cage and pour the flies into the 5-L beaker.(7)Flies that fall on the towels can now be thrown into the beaker as well.(8)Gas the old cage again to blow loose the remaining living flies and shake the cage over the beaker.(9)Pour the flies from the beaker into the new cage.(10)Close the freshly filled cage, bring it to its designed place in the lab and supply the flies with food.(11)Clean the old cage and the nylon with tap water. Do not use soap or bleach. Rinse with distilled water.

### 3.2. Egg Collection and Preparation for Microinjections

*D. guttifera* females lay comparably few eggs within a given period. For this reason, the cage setup described above is used to obtain at least 200–300 freshly fertilized eggs per hour from the population of ~10,000 mixed-sex flies. Keep the cage on the lab bench at room temperature and normal lab humidity with a 40-W lamp placed ~30 cm above it, which is timed to turn on at 6 a.m. and off at 6 p.m. In the cage, females need to be reared on fresh food with dry yeast at all times in order for their ovaries to develop and to achieve a continuous egg production. The age of the females varies, as fresh flies are provided from the bottle cultures every week. If the food is in poor condition, females will retain their fertilized eggs inside of their bodies and lay them when the eggs are too old for microinjections. Supply the cage with fresh cornmeal-sucrose-yeast medium 3–4 times a week and especially one day before injections are carried out.

Eggs that are laid in the morning are often already cellularized and tend to have a thicker chorion, which is disadvantageous for injections. The egg quality increases later during the day. The best time to start the egg-laying is 11 a.m. Harvest the eggs in 1-h cycles, starting at noon. It is advisable that one person lines up two slides (~200 eggs) during a 1-h cycle, while another person injects them. All wash steps of the eggs are done with distilled water.


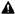
**CRITICAL STEPS:** Never wash *D. guttifera* eggs with ethanol, because it seems to harden the chorion and prevents the injection needles from penetrating it, and never leave the cage without food.

Make a starter yeast paste from instant baker’s yeast grains one day before using it, and on the morning of injection, mix it again with additional dry yeast and water. Keep the yeast paste at room temperature during the injections; it should smell a bit like fermented fruit. At 11 a.m., replace all cornmeal-sucrose-yeast medium plates in the cage with four moist sponges, which are to be placed into medium-sized (100 mm × 15 mm) Petri dish bottom halves and covered with a thin layer of fresh, active yeast paste (Figure 2). A second set of four sponges will be needed for the next round.

The best way to prepare the sponges is to first make them too wet and then to carefully squeeze off the excess water with two thumbs pressing and moving from the top to the bottom throughout the surface of the sponges, until no drops are running out easily. The sponges should be dry enough that no water runs out when flies are bumped off from them during removal from the cage but wet enough to supply the flies with enough liquid. Sponges that are too dry seem to result in thicker coronas of the eggs (the thickening near the entry site of the injection needle), while sponges that are too wet cause the accumulation of liquid in the cage. The yeast paste is then supplied evenly over the sponge surface. Cover the sponges with lids before placing them into the cage to prevent stray flies from laying eggs on them.

Allow the females to lay eggs on the 4 sponges for one hour. The flies prefer to sit on the sponge part that is closest to the cage wall. Sponges that are placed too far away from the wall are often ignored by the flies.

To collect the eggs, remove the sponges one by one from the cage and immediately replace them with sponges covered with fresh yeast paste. Gently squeeze out the eggs-containing sponges one by one in a 2-L jar filled with 1 L of distilled water (repeatedly bend the sponges from the edge outward to avoid crushing the eggs, and the eggs will drop out of the pores), then allow the eggs to sink to the bottom of the 2-L jar for 2 min. While waiting, wash the sponges and Petri dishes with distilled water. The sponges should be squeezed hard while washing to kill any possible remaining eggs. After the four used sponges are washed, put fresh yeast paste on them and store them in a drawer (away from contaminating flies) until they are used again. At the end of the injection day, wash the sponges thoroughly and place them on a few paper towels on the bench for drying.

Assemble—from top to bottom—a funnel, a coarse-filter basket, and a fine-filter basket (Figure 3A). Next, pour the upper half of the egg collection water into the sink (the eggs are at the bottom of the jar). Then swirl and pour the remaining water with the eggs into the funnel/filter system. The coarse-filter traps sponge fibers and flies, while the fine filter collects the eggs (Figure 3B). Rinse the 2-L jar once with 100–200 mL of distilled water, then pour it through the filter system to collect the remaining eggs. With a squeeze bottle of distilled water, wash the eggs off the upper filter and allow them to slip through the coarse mesh into the fine filter below. After obtaining a sufficient number of eggs in the lower filter, remove the baskets from the funnel and wash the eggs thoroughly with distilled water from a squeeze bottle. Then, put the basket with the washed eggs into a bottom half of a small Petri dish, which is halfway filled with distilled water. Line up the eggs under a dissection scope.

### 3.3. Lining up the Eggs for Microinjections

Place a micro cover glass (18 mm × 18 mm) onto a microslide (25 mm × 75 mm, 1 mm thick) with a small droplet of water so that it sticks. Dry off the edges with a piece of tissue. The washed eggs are now transferred from the basket onto the cover glass as a clump, using a small brush. It is essential that the eggs are kept wet at all times. During lining up, always pick only the white and structureless-appearing eggs. It is advisable to put a clump of more than 100 eggs into the upper middle of the cover glass. Then, use a very fine brown-haired brush, make it wet, briefly dry it on a tissue paper, pull the first egg into position, dry the brush, pull the next egg to extend the line, and so on.

Align the first 10 eggs with a relatively dry (very fine) brush, or otherwise, the surface tension will make it impossible to position them correctly. Then, gently move the clump of ~100 eggs down while extending the line of selected eggs downwards, otherwise you lose sight of your egg clump, and the eggs may dry out and die by accident. In the end, the row consists of about 100 eggs, which shall have their posterior end towards the right edge of the cover glass. The row of eggs should be positioned about 5 mm away from that edge. Keep the eggs moist at all times by supplying water with the brush to the clump of unaligned eggs, as well as the top, middle, and end of the row of aligned eggs.

When the row is completed, remove all unused eggs and let the row dry, while carefully monitoring the eggs under the microscope. The row can now be rearranged slightly U-shaped to make them fit better to the cornmeal-sucrose-yeast medium surface later. When the egg line appears completely dry, wait about 3–5 more seconds (depending on the humidity of the room) and then quickly cover them with the Halocarbon oil mixture (7 parts of type 700 + 1 part of type 27), supplied by a syringe (Figure 4). The Halocarbon oil stops further desiccation but allows breathing. The halocarbon oil mix must be prepared at least a day before you can use it, so plan ahead.


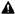
**CRITICAL STEPS:** If the eggs become too desiccated (oil added too late), they die and become deformed when the needle pokes against them. If, however, the oil is put on too early, the eggs will not stick to the glass and move away when the injection needle is poked against them. Only a few seconds lay between both extremes. Do not try to inject a slide with swimming or dead eggs. Instead, save your needles for a better slide.
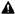
**CRITICAL STEPS:** Note the following important timelines: egg-laying in the cage = 1 h, egg collection and lining up = 20 min, injecting one slide containing about 100 eggs = 15 min. It is important to ensure that the eggs are in the syncytial blastoderm stage during injection, which lasts for about 2 h or less at room temperature.

### 3.4. Preparation of Transgene DNA

(1)Follow the manufacturer’s instructions on the isolation of the plasmid DNA containing the transgene of interest from an overnight *Escherichia coli* culture. In this described experiment, we use the Invitrogen HiPure Plasmid Midi-Prep Kit by Thermo Fisher Scientific (Waltham, MA, USA).(2)Add 3.5 mL of isopropanol to the eluted DNA in a Corex glass vial.(3)Close the Corex vial with Parafilm, invert 10 times, take away the Parafilm, and wipe off any drops with Kimwipes.(4)Centrifuge at 11,000 rpm for 30 min at 4 °C.(5)Mark the area of the DNA pellet with a marker, decant, and invert the vial on a paper towel.(6)Add 350 µL of sterile MQ water and dissolve the DNA pellet by vortexing.(7)Collect the plasmid DNA in a 1.5-mL Eppendorf tube.(8)Add 35 µL of 3 M of sodium acetate (pH 5.5).(9)Add 875 µL of 200-proof ethanol and invert 10 times.(10)Spin the DNA at 14,000 rpm for 20 min at 4 °C on a bench-top centrifuge.(11)Remove supernatant and wash with 300 µL of 70% ethanol by inverting the tube 10 times.(12)Centrifuge again at 14,000 rpm for 10 min at room temperature.(13)Remove supernatant and centrifuge for 1 min.(14)Remove all ethanol and dry the pellet for 5 min.(15)Dissolve the DNA in 50–100 µL of elution buffer, measure the DNA concentration, and add more elution buffer if necessary, to a final concentration of 1 µg/µL.

### 3.5. Preparation of the Injection Cocktail

(1)On ice, add 20 µL of transgene-containing *piggyBac* construct (1 µg/µL) to a 1.5-mL test tube, 5 µL of the *piggyBac* helper plasmid (phspBac ([1 µg/µL)), and 15 µL of sterile distilled water. Note: We do not add food color, because it is not necessary for the visualization of the injected material.(2)Centrifuge the DNA mixture for 20 min at maximum speed, and transfer the upper 39 µL of it to a fresh tube (do not touch the bottom of the tube with the pipette tip to prevent transferring undissolved material that would clog the needle). Spin the DNA again for 20 min and transfer 38 µL of the injection mix into another fresh tube. This DNA is now clear of debris and can be used to load the needles. The *piggyBac* system pBac (backbone)/phspBac (*piggyBac* helper) results in 1 transformation event every 50 *D. guttifera* eggs with an empty vector and 1 every 500 eggs with an 8-kb insert [18].

### 3.6. Needle Preparation and Microinjections

Use a Flaming/Brown micropipette puller Model P-97 and FHC capillary tubing (Borosil 1.0 × 0.75 mm ID/Fiber with Omega dot fiber) to obtain the standard needles. We currently use the following parameters, but these parameters can change based on the filament, machine, and humidity of the room: P = 500, Heat = 496, Pull = 125, Vel = 10, Time = 186 [19]. Do not use gloves when pulling needles, but wash your hands before touching the capillaries! Extreme care should be taken to avoid touching the platinum heating filament of the machine with the glass capillaries. Load the needles at least one hour before using them with approximately 0.5 µL of the DNA mix, and store them in a moist chamber at 4 °C for a few days. Use an automated injection system based on nitrogen pressure, a micromanipulator for holding the needle, and an inverted microscope for viewing.

The very tip of every needle must first gently be broken off before the first injection can happen. To do this, carefully touch the chorion of an egg with the needle tip, while continuously pushing the "BALN" or “INJ” button (Appendix A), until the DNA flows out. (Different models may have different button names; please refer to the user manual.) Ensure that the needle is not badly broken, as this may cause the granular cytoplasmic content to ooze out of the egg (Figure 5A). Inject the eggs with a very fine needle tip into the posterior-most part (Figure 5B) by hitting either the “INJ” button or the foot pedal (it is a matter of preference).

In order to prevent the needle from getting clogged, the “BALN” or “INJ” button or the foot pedal should be hit frequently, while the needle tip is outside of the eggs. If the needle is clogged, carefully rub the tip of the needle against an egg while holding the BALN button pressed. Additionally, moving the eggs away from the needle at high speed can help make the DNA flow again.


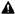
**CRITICAL STEP:** The needles must be changed when the eggs start leaking out a lot of granular cytoplasm, whereas small clear droplets are acceptable and normal.

### 3.7. Post-Injection Treatment and Heat Shock of the Eggs

After injecting all eggs on a slide, remove the slide from the microscope stage, remove the cover glass from the slide, and briefly let the oil drop off. Quickly wipe off some excess oil against the microscope slide. Stick the cover glass twice into a fresh cornmeal-sucrose-yeast medium vial, thereby carving out a small trench of food. Then place the posterior ends of the eggs with the cover slip into the trench without pushing the eggs too hard against the food. Close the vial with a cotton ball and squeeze enough distilled water into the cotton ball to provide immediate moisture (do not make them dripping-wet but sufficiently moist). Place the vial vertically into a wet chamber (a beaker with 5-mm-high water on the ground and 2 wet paper towels stuck against the inner wall). Hold up to 8 vials with injected eggs together with a rubber band (Figure 6). Avoid water flowing into the vials by keeping the cotton balls low in the vials, so that they do not touch the wet paper on the side of the wet chamber. Close the chamber with aluminum foil, label the lid, and store it at room temperature.

On the next morning (16–18 h after injection), heat shock the eggs at 40 °C for 90 min in the water bath (use a rubber band to hold the vials together, and submerge the vials as deep as needed to get the food level under water; do not let any water get into the vials!). After the heat shock, put the vials back into the chamber (now remove the wet paper towels that lined the wall). Place the foil lid back on, and store the chamber (with some water at the bottom) at room temperature for roughly a week. Check daily for problems (mold or flooded vials) and the appearance of the first pupae.

### 3.8. After-Care and Fly Crosses

Look at the moist chamber every day to ensure it remains moistened. When most pupae have formed and no more wandering larvae are seen, collect the pupae from the cotton plugs with a brush and two pairs of forceps (Figure 7A). Making the cotton balls very wet helps to easily detach the pupae. Do not leave much cotton left on the pupae; however, a few fibers can remain attached. Collect the pupae temporarily on a wet tissue paper. Take a clean and empty glass vial ("hatching vial"), place a long piece of anti-fungus paper along one side of the wall, and moisten the paper with a squirt bottle (remove excess water with a paper towel afterwards so that no free water is in the vial, or else the pupae will drown). Transfer the collected pupae with a moist brush from the wet tissue paper onto the anti-fungus paper in the vial. Then, collect the remaining pupae of the same construct as described above, and combine them with the others in the same hatching vial. Close the hatching vial with a fresh cotton ball (push the ball deep enough down to prevent it from picking up water), and store it angled upwards (on a small plastic lid from a pipette box) in a moist chamber (Figure 7B).


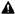
**CRITICAL STEP:** Moisten the anti-fungus paper inside the glass vials whenever needed, and never let the anti-fungal paper dry out. Check the chamber twice daily.

Start collecting wild-type virgin flies of both sexes for backcrosses as soon as you see the first pupae forming from the injection experiment vials. The collected virgins are stored in cornmeal-sucrose-yeast medium vials with a few grains of dry baker’s yeast and must be bumped to fresh vials every 3–4 days (or they will get stuck in slimy food). They will be sexually mature when your first injection survivors hatch. The hatched flies from the injections (the injection survivors) are collected once a day in the late afternoon on a CO_2_ pad and immediately crossed with wild-type virgins of the opposite sex. Use cornmeal-sucrose-yeast medium vials with a few grains of baker’s yeast, and combine 3 males or 10 females of the injection survivors with 15 wild-type virgins of the opposite sex in one vial. The lumping of injection survivors into the same cross is done to produce enough larvae per vial (to keep microorganisms in the culture down). Bump the crosses to fresh food every 3–4 days. Look at your crosses and their offspring every day. Screen for transgenic larvae 3 days after the parents have been removed from the vials.

### 3.9. Screening for Transgenic Larvae

The larval enhanced green fluorescent protein (EGFP) screen is carried out in the fly room with the main lights switched off as follows:(1)Switch on the UV lamp 10 min prior to screening.(2)Microscope settings: light filter = “GFP”, bottom filter = “Oblique”, objective = 1× shutter open, camera tract closed, and all visible microscope lights off (check that the bottom light is off as well!).(3)Repeat the following steps until all vials have been screened:
(a)Remove the larvae from a food vial by squirting distilled water (squeeze bottle) and stirring with a rough brush; then, pour the larva/food soup into a medium-sized Petri dish bottom half.(b)Collect positive larvae (Figure 8) with a pair of forceps into 2-mL tubes filled with 1-mL cornmeal-sucrose-yeast medium (briefly angle the food by centrifugation, allowing the larvae to be removed more easily from the collection forceps). Wash and screen each vial 3 times for transgenic larvae.(c)Poke two holes into each lid (not too big that the larvae can escape) after putting larvae into the 2-mL food tubes (only add up to 10 larvae per vial). Label each vial with the construct and line information.(d)Discard screening soup with negative larvae and food debris into an empty 5-L beaker. Rinse the Petri dish with distilled water after screening each vial.


## 4. Expected Results

The perfect timepoint for screening for transgenic larvae is 3 days after the parents have been removed from the vials, as all larvae would have hatched, but the pupae would not have formed yet. EGFP can only be detected in larvae but not in eggs, pupae, or the adult flies. Look for EGFP in the larval eye disks, Bolwig organs, brain, and/or in the anal plates (Figure 8) [20]. EGFP expression is usually strong and clearly visible in all larval instars. Finding multiple positive larvae per vial is not uncommon. Note that many transgenic lines are obtained in later screening sessions, so it is worth the time and patience bumping the crosses for 2 or 3 weeks before declaring them negative.

## 5. Reagents

### 5.1. Anti-Fungus Paper 

Dissolve 2 g of sorbic acid and 0.6 g of methyl paraben in 200 mL of 95% ethanol. Roll ~20 paper towels and stick them into a 1-L glass beaker. Slowly pour the anti-fungus solution over the towels to soak them evenly. Unfold the towels and let them dry completely. Anti-fungus paper does not go bad and can be stored for years.

### 5.2. Halocarbon Oil Mixture

Use a 50-mL Falcon tube, add 35 mL of Halocarbon oil 700 and 5 mL of Halocarbon oil 27. Homogenize the mixture on a nutator overnight. 

## 6. Conclusions

The starting point for this protocol was the standard method that was optimized for *D. melanogaster*. We quickly learned that “whatever works for *D. melanogaster* does not work for/kills *D. guttifera*.” It took therefore several years of innovations and deviations from the original protocol to make *D. guttifera* survive the procedure from the beginning to the end and to obtain the first transgenic animals. Key innovations are (1) establishing a large cage population (this compensates for the low egg-lay rate), (2) having good cage aeration (this prevents the accumulation of ethanol fumes, which become quickly lethal for ethanol-susceptible species), (3) using sponges as egg-lay surfaces (this allows for easy egg collections, even when females tend to stick their eggs into substrates), (4) no ethanol wash step to visualize the interior of the eggs before injection (this prevents the egg chorion from becoming rock-hard), and (5) washing larvae out of the food for transgenic screening (no need for a white-eyed mutant). We believe that our method, developed for a very delicate species, should be useful to transform most drosophilid species that can be reared in the laboratory on a standard fly-food medium, such as *Drosophila tripunctata*, *Drosophila quinaria*, *Drosophila subquinaria*, and *Drosophila deflecta*. For recommendations on collecting and rearing many non-model species, please refer to our book [10].

## Figures and Tables

**Figure 1 mps-03-00031-f001:**
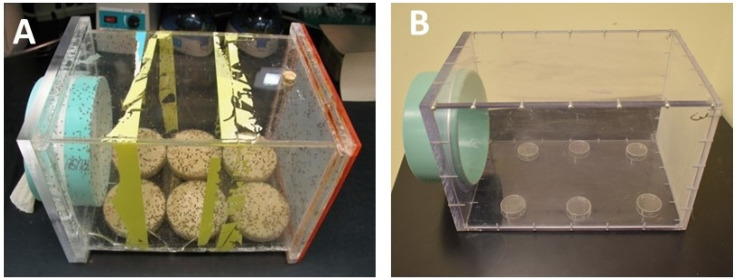
The egg-laying cage measuring 300 mm × 200 mm × 200 mm. (**A**) An egg-laying cage holding ~10,000 flies, which contains six plates of cornmeal-sucrose-yeast medium. (**B**) An empty egg-laying cage showing the six small (35 mm × 10 mm) plastic Petri dish bottom halves taped upside-down onto the bottom of the cage using double-sided tape.

**Figure 2 mps-03-00031-f002:**
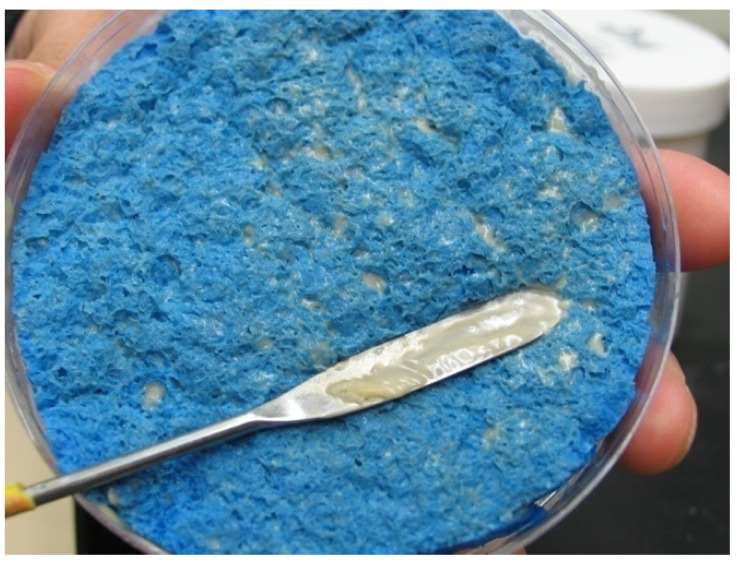
The yeast paste is evenly applied with a flat spatula over the sponge surface.

**Figure 3 mps-03-00031-f003:**
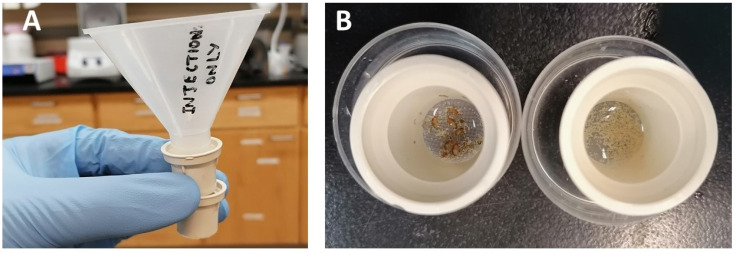
Egg collection with a funnel and two filters. (**A**) The funnel is placed on top of the coarse filter, and the fine filter is held underneath to collect the eggs. (**B**) The coarse filter (left) mainly traps the flies and sponge fibers, while the fine filter (right) retains the eggs.

**Figure 4 mps-03-00031-f004:**
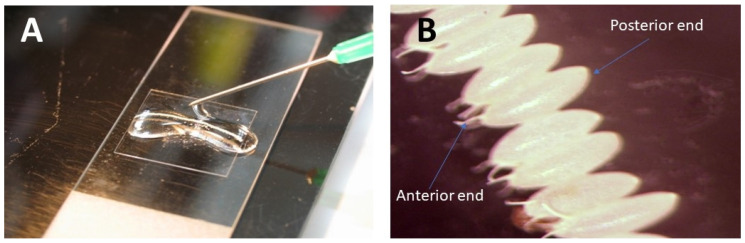
(**A**) Adding the halocarbon oil mixture to a line of eggs. (**B**) The anterior ends of the eggs contain the filaments (left), while the posterior ends point to the right.

**Figure 5 mps-03-00031-f005:**
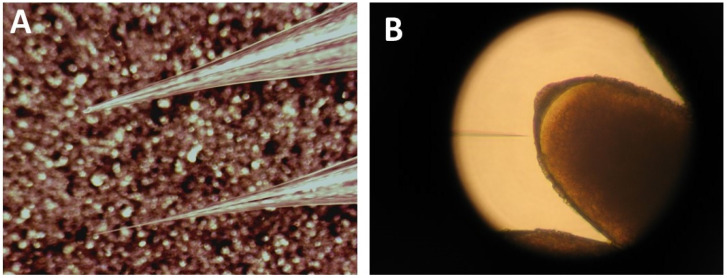
Needle qualities and correct injection site. (**A**) The top needle is a badly broken needle, while the second needle is the standard needle. Note that the best needle is the one that has a beveled tip with a small opening. (**B**) The transgene can be injected through the posterior part of the egg.

**Figure 6 mps-03-00031-f006:**
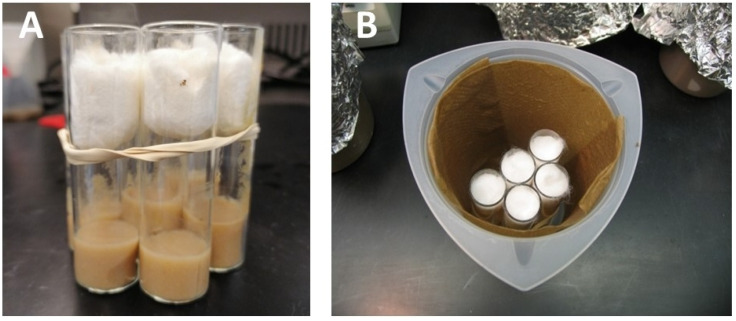
(**A**) Vials are bundled together with a rubber band. (**B**) A wet chamber for the vials containing injected eggs.

**Figure 7 mps-03-00031-f007:**
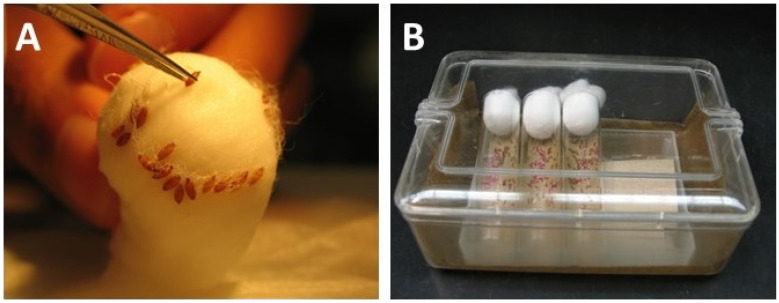
Collection of pupae and after-care. (**A**) Pupae are collected from the cotton plugs with fine forceps. (**B**) Pupae collected on moist anti-fungus paper in a hatching vial and stored angled upwards in a moist chamber.

**Figure 8 mps-03-00031-f008:**
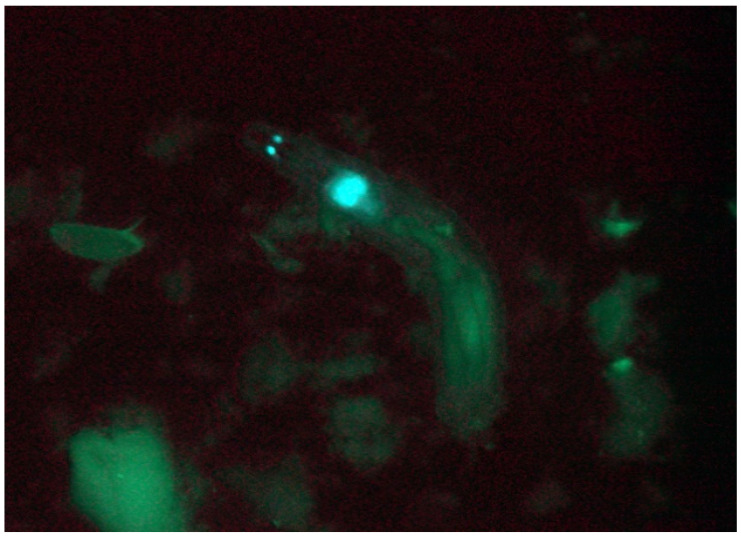
A transgenic *D. guttifera* larva showing enhanced green fluorescent protein (EGFP) expression in the Bolwig organs (two small glowing dots on the left) and the eye disks/brain (larger blob to the lower right).

**Table 1 mps-03-00031-t001:** Troubleshooting guide (summarizes a variety of problems that we have encountered in the past. The table explains the reasons underlying the problems and offers appropriate actions to work around each problem).

Problem	Reason	Solution
Few larvae hatch.	Bad injection needles.	Make sure you use standard needles. Strictly follow the instructions under Section 3.6.
	Over-desiccation of eggs during lining up.	Add injection oil within 3–5 s after eggs appear completely dry under the microscope.
	Toxic DNA.	Use highly pure DNA (no mini-preps). Additional isopropanol precipitation after the first extraction of DNA is necessary.
	Too-high heat shock temperature.	The heat shock temperature for D. guttifera transformation is 40 °C; however, if this temperature is causing strong reduction in the survival rate, reduce the temperature by 1–2 degrees but never go below 37 °C.
	If the Halocarbon oil mixture is suffocating and killing the eggs. This does not normally happen, but we have seen this problem in our lab.	Use Halocarbon oil 27 only or GEM extra virgin olive oil. We are using extra virgin olive oil in our lab with high success.
Few surviving adults.	Having only a few injection-surviving larvae causes bacterial growth and/or mold formation in the food.	Ensure a high injection survival rate. At least 10–20 larvae can keep the food mold-free.
	Desiccation of pupae due to dry chamber and/or dry cotton plugs.	Always keep the chamber and the cotton plugs
No transgenic larvae.	Poor DNA quality.	Always clean the DNA by isopropanol precipitation or prepare new DNA samples.
	Insufficient heat shock temperature.	Always ensure the heat shock temperature is not below 37 °C. Check the temperature of your water bath with a calibrated thermometer.

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
