# Peer review of "The Making of Transgenic Drosophila guttifera"

_mps, 2020, doi:10.3390/mps3020031_

Round 1

Reviewer 1 Report

The authors present methods for transforming Drosophila gutiffera.  Generally the work appears to be a step forward and there seem to be differences from publications on other organisms.  However, there are some details that would help in understanding the presentation and some context that could be added after method description and initial results.

1.  CO2 / 5L Beaker, etc are mentioned in methods, but do not seem to be in equipment list.  Go back through and align these parts of the section.

2. What is meant by "bump"

3. Figure 3: Canyou show a macro view of the 2L transfer of embryos from the side?

4. Withheld eggs are an issue for other species as well.  D. sechellia comes to mind. Worth noting in a discussion?  How many other species could be more effectively transformed if your alterations (or specific alterations) to current standards are implemented?

5. When you note not to touch bottom of DNA tube, can you clarify why?

6. Figure 6 Can you show a side view of a vial too?

7.  With D. mel, eggs can be left in halocarbon oil on the slide (slide placed on a plate of food) until hatching.  Why not here?

8. Intro/conclusion work: It would really help to explicitly note the key innovations this group has implemented over previous protocols.  What they are and why implemented/problems they solve.  In the conclusion it may also be worth noting the types of species that may also be more effectively transformed with some of these alterations besides D. gutiffera (the D. sechellia comparison I noted above - if relevant - might apply.  Surely there are other species with similar challenges for at least part of the protocol).  While the method is good for one novel drosophilid, there may be other species where similar alterations to standard approaches would help.

Reviewer 2 Report

In the presented manuscript, Shittu et al. Describe a protocol for performing transgenics in Drosophila guttifera, a particularly difficult species to work with in wet lab experiments. The authors thoroughly outline how to best collect an appropriate number of eggs for transgenics, how to treat these delicate eggs so they are best prepared for injections, and how to care for a screen the resulting transformed eggs.

I have a few minor criticisms and questions regarding the wording of certain steps in the protocol, but otherwise I feel the authors have done a good job developing and describing a protocol for experimental manipulation of non-model Drosophila. I feel the manuscript could do with a careful reread to change several minor mistakes and fix sentences that are confusingly worded or do not flow.

Line 16-19: The sentences “methods describing transgenesis…” “However, these methods...” Could be combined and cut down and would flow much better.

Line 28: ‘Our ability to genetically manipulate any organism…’ reads better.

Line 37: Add a sentence to the end here about why manipulation in D. mel doesn’t mean its useful for everybody.

Line 38: “Drosophila guttifera is a rare mycophageous species in the quinarian species group, native to the Midwest and southeast of the usa” reads better.

Line 48: “Previously, a review highlighting strategies for making transgenic D. melanogaster was published” reads better

Line 63: swap ‘insect enthusiasts’ for ‘researchers’

Line 159: Could the authors spell out how many flies are needed for 200-300 eggs per hour? The previous section implies 1000, but knowing the exact number of males and females would be helpful.

Line 189: The wording of this sentence is confusing.

Line 206: There is no figure 3A and 3B, either fix the text or the figures. I assume figure 3B is missing.

Line 213: Here the authors describe soaking the eggs in water, then say to line up the embryos under a dissection scope. It implies the embryos are different from the eggs. Maybe remove this last sentence as it contradicts the next paragraph also. The authors should take care to refer to the eggs as either embryos or eggs, as this last sentence and the next paragraph confused me due to the switching back and forth.

Line 217: I would appreciate a figure here showing the eggs lined up on the slide, as no reference point is given to know what is the anterior or posterior. I assume the side with the filaments in the anterior, but I would appreciate if this was clarified.

Line 254: Swap ‘in our lab’ for ‘in this described experiment’

Line 290: ‘For up to a few days’ How many is a few days? Either say ‘For a few days’ or ‘For X days’.

Line 292: Maybe a supplementary image of the described microinjector so that people using a different brand will have a reference point for what ‘BALN’ and ‘INJ’ refer to on their injector.

Line 325: How long are the vials submerged in cold water, the text implies for a week, though they are also at room temperature for a week.

Line 381: Are the transgenic constructs expected to be visible in the adults as well? Clarification would be appreciated.

Reviewer 3 Report

Summary:

The authors present a detailed protocol describing an approach to generate transgenic Drosophila guttifera. The protocol itself is clearly written, and the authors have provided a troubleshooting table which is a helpful summary of the key important points of the protocol. Having never performed injections myself, I feel that I would be able to successfully carry out this protocol. My suggestions on this protocol are minor, but I do feel that they would make a few points even clearer.

Lines 44-45: The statement summarizing the pupal development study could be elaborated on a bit. As is, it really doesn’t add much to the introduction.

Lines 45-47: This sentence makes it sound like the papers just discussed have accomplished transgenics with D. guttifera. Is this an accurate interpretation?

Line 57: Point 6 suggests it is already known that the piggyBac transposon works in D. guttifera. Please provide a citation for this information.

Line 61: The mentioned video (or video link placeholder) is missing.

Line 64: unnecessary comma after enthusiasts.

Line 94: Please provide an item number for the anti-fungus paper

Line 116-117: Should the statement be reversed? Wipe off the plates and then sprinkle with yeast?

Line 140-141: Clarify that this is to prevent CO2 leaking out the cage through the back wall.

Figure 1 caption, line 157: unnecessary comma after cage.

Line 158: Please provide guidelines for temperatures, humidity, and age of females used to produce embryos for collection in this section.

Line 204-206: The sentence referencing Figure 3A is choppy. Consider revising to discuss the filter setup, then describing how the embryos are filtered through.

Line 209: Possibly a typo—“wash the eggs off” instead of “of”

Figure 3: Figure 3B is missing

Line 335-336: Consider revising the statement about the sticky pupae.

Round 2

Reviewer 1 Report

Authors have done a good job of replying to reviewers.  A few examples of other Drosophila that would benefit from specific modifications would help, but generally they did a good job. 
